# On Learning Intrinsic Rewards for
# Policy Gradient Methods

**Zeyu Zheng**                    **Junhyuk Oh**                    **Satinder Singh**
Computer Science & Engineering
University of Michigan
{zeyu,junhyuk,baveja}@umich.edu

## Abstract

In many sequential decision making tasks, it is challenging to design reward
functions that help an RL agent efficiently learn behavior that is considered good
by the agent designer. A number of different formulations of the reward-design
problem have been proposed in the literature. In this paper we build on the Optimal
Rewards Framework of Singh et al. [2010] that defines the optimal intrinsic reward
function as one that when used by an RL agent achieves behavior that optimizes the
task-specifying or extrinsic reward function. Previous work in this framework has
shown how good intrinsic reward functions can be learned for lookahead search
based planning agents. Whether it is possible to learn intrinsic reward functions
for learning agents remains an open problem. In this paper we derive a novel
algorithm for learning intrinsic rewards for policy-gradient based *learning* agents.
We compare the performance of an augmented agent that uses our algorithm to
provide additive intrinsic rewards to an A2C-based policy learner (for Atari games)
and a PPO-based policy learner (for Mujoco domains) with a baseline agent that
uses the same policy learners but with only extrinsic rewards. We also compare our
method with using a constant "live bonus" and with using a count-based exploration
bonus (i.e., pixel-SimHash). Our results show improved performance on most but
not all of the domains.

## 1 Introduction

One of the challenges facing an agent-designer in formulating a sequential decision making task
as a Reinforcement Learning (RL) problem is that of defining a reward function. In some cases a
choice of reward function is clear from the designer's understanding of the task. For example, in
board games such as Chess or Go the notion of win/loss/draw comes with the game definition, and in
Atari games there is a game score that is part of the game. In other cases there may not be any clear
choice of reward function. For example, in domains in which the agent is interacting with humans in
the environment and the objective is to maximize human-satisfaction it can be hard to define a reward
function. Similarly, when the task objective contains multiple criteria such as minimizing energy
consumption and maximizing throughput and minimizing latency, it is not clear how to combine
these into a single scalar-valued reward function.

Even when a reward function can be defined, it is not unique in the sense that certain transformations
of the reward function, e.g., adding a potential-based reward [Ng et al., 1999], will not change
the resulting ordering over agent behaviors. While the choice of potential-based or other (policy)
order-preserving reward function used to transform the original reward function does not change what
the optimal policy is, it can change for better or for worse the sample (and computational) complexity
of the RL agent learning from experience in its environment using the transformed reward function.

Yet another aspect to the challenge of reward-design stems from the observation that in many complex real-world tasks an RL agent is simply not going to learn an optimal policy because of various bounds (or limitations) on the agent-environment interaction (e.g., inadequate memory, representational capacity, computation, training data, etc.). Thus, in addressing the reward-design problem one may want to consider transformations of the task-specifying reward function that change the optimal policy. This is because it could result in the bounded-agent achieving a more desirable (to the agent designer) policy than otherwise. This is often done in the form of shaping reward functions that are less sparse than an original reward function and so lead to faster learning of a good policy even if it in principle changes what the theoretically optimal policy might be [Rajeswaran et al., 2017]. Other examples of transforming the reward function to aid learning in RL agents is the use of exploration bonuses, e.g., count-based reward bonuses for agents that encourage experiencing infrequently visited states [Bellemare et al., 2016, Ostrovski et al., 2017, Tang et al., 2017].

The above challenges make reward-design difficult, error-prone, and typically an iterative process. Reward functions that *seem* to capture the designer's objective can sometimes lead to unexpected and undesired behaviors. Phenomena such as reward-hacking [Amodei et al., 2016] illustrate this vividly. There are many formulations and resulting approaches to the problem of reward-design including preference elicitation, inverse RL, intrinsically motivated RL, optimal rewards, potential-based shaping rewards, more general reward shaping, and mechanism design; often the details of the formulation depends on the class of RL domains being addressed. In this paper we build on the optimal rewards problem formulation of Singh et al. [2010]. We discuss the optimal rewards framework as well as some other approaches for learning intrinsic rewards in Section 2.

Our main contribution in this paper is the derivation of a new stochastic-gradient-based method for learning parametric *intrinsic* rewards that when added to the task-specifying (hereafter *extrinsic*) rewards can improve the performance of policy-gradient based learning methods for solving RL problems. The policy-gradient updates the policy parameters to optimize the sum of the extrinsic and intrinsic rewards, while simultaneously our method updates the intrinsic reward parameters to optimize the extrinsic rewards achieved by the policy. We evaluate our method on several Atari games with a state of the art A2C (Advantage Actor-Critic) [Mnih et al., 2016] agent as well as on a few Mujoco domains with a similarly state of the art PPO agent and show that learning intrinsic rewards can outperform using just extrinsic reward as well as using a combination of extrinsic reward and a constant "live bonus" [Duan et al., 2016a]. On Atari games, we also compared our method with a count-based methods, i.e., pixel-SimHash [Tang et al., 2017]. Our method showed better performance.

## 2   Background and Related Work

**Optimal rewards and reward design.**   Our work builds on the Optimal Reward Framework [Singh et al., 2010]. Formally, the optimal intrinsic reward for a specific combination of RL agent and environment is defined as the reward that when used by the agent for its learning in its environment maximizes the extrinsic reward. The main intuition is that in practice all RL agents are bounded (computationally, representationally, in terms of data availability, etc.) and the optimal intrinsic reward can help mitigate these bounds. Computing the optimal reward remains a big challenge, of course. The paper introducing the framework used exhaustive search over a space of intrinsic reward functions and thus does not scale. Sorg et al. [2010] introduced PGRD (Policy Gradient for Reward Design), a scalable algorithm that only works with lookahead-search (e.g., UCT) based planning agents (and hence the agent itself is not a learning-based agent; only the reward to use with the fixed planner is learned). Its insight was that the intrinsic reward can be treated as a parameter that influences the outcome of the planning process and thus can be trained via gradient ascent as long as the planning process is differentiable (which UCT and related algorithms are). Guo et al. [2016] extended the scalability of PGRD to high-dimensional image inputs in Atari 2600 games and used the intrinsic reward as a reward bonus to improve the performance of the Monte Carlo Tree Search algorithm using the Atari emulator as a model for the planning. A big open challenge is deriving a sound algorithm for learning intrinsic rewards for learning-based RL agents and showing that it can learn intrinsic rewards fast enough to beneficially influence the online performance of the learning based RL agent. Our main contribution in this paper is to answer this challenge.

**Reward shaping and Auxiliary rewards.**   Reward shaping [Ng et al., 1999] provides a general answer to what space of reward function modifications do not change the optimal policy, specifically potential-based rewards. Other attempts have been made to design auxiliary rewards with desired properties. For example, the UNREAL agent [Jaderberg et al., 2016] used pseudo-reward computed from unsupervised auxiliary tasks to refine its internal representations. In Bellemare et al. [2016], Ostrovski et al. [2017], and Tang et al. [2017], a pseudo-count based reward bonus was given to the agent to encourage exploration. Pathak et al. [2017] used self-supervised prediction errors as intrinsic rewards to help the agent explore. In these and other similar examples [Schmidhuber, 2010, Stadie et al., 2015, Oudeyer and Kaplan, 2009], the agent's learning performance improves through the reward transformations, but the reward transformations are expert-designed and not learned. The main departure point in this paper is that we learn the parameters of an intrinsic reward function that maps high-dimensional observations and actions to rewards.

**Hierarchical RL.**   Another approach to a form of intrinsic reward is in the work on hierarchical RL. For example, FeUdal Networks (FuNs) [Vezhnevets et al., 2017] is a hierarchical architecture with a Manager and a Worker learning at different time scales. The Manager conveys abstract goals to the Worker and the Worker optimizes its policy to maximize the extrinsic reward and the cosine distance to the goal. The Manager optimizes its proposed goals to guide the Worker to learn a better policy in terms of the cumulative extrinsic reward. A large body of work on hierarchical RL also generally involves a higher level module choosing goals for lower level modules. All of this work can be viewed as a special case of creating intrinsic rewards within a multi-module agent architecture. One special aspect of hierarchical-RL work is that these intrinsic rewards are usually associated with goals of achievement, i.e., achieving a specific goal state while in our setting the intrinsic reward functions are general mappings from observation-action pairs to rewards. Another special aspect is that most evaluations of hierarchical RL show a benefit in the transfer setting with typically worse performance on early tasks while the manager is learning and better performance on later tasks once the manager has learned. In our setting we take on the challenge of showing that learning and using intrinsic rewards can help the RL agent perform better while it is learning on a single task. Finally, another difference is that hierarchical RL typically treats the lower-level learner as a black box while we train the intrinsic reward using gradients through the policy module in our architecture.

**Meta Learning for RL.**   Our work can be viewed as an instance of meta learning [Andrychowicz et al., 2016, Santoro et al., 2016, Nichol and Schulman, 2018] in the sense that the intrinsic reward function module acts as a meta-learner that learns to improve the agent's objective (i.e., mixture of extrinsic and intrinsic reward) by taking into account how each gradient step of the agent affects the true objective (i.e., extrinsic reward) through the meta-gradient. However, a key distinction from the prior work on meta learning for RL [Finn et al., 2017, Duan et al., 2017, Wang et al., 2016, Duan et al., 2016b] is that our method aims to meta-learn intrinsic rewards within a single task, whereas much of the prior work is designed to quickly adapt to new tasks in a few-shot learning scenario. Xu et al. [2018] concurrently proposed a similar idea that learns to find meta-parameters (e.g., discount factor) such that the agent can learn more efficiently within a single task. In contrast to state-independent meta-parameters in [Xu et al., 2018], we propose a richer form of state-dependent meta-learner (i.e., intrinsic rewards) that directly changes the reward function of the agent, which can be potentially extended to hierarchical RL.

## 3   Gradient-Based Learning of Intrinsic Rewards: A Derivation

As noted earlier, the most practical previous work in learning intrinsic rewards using the Optimal Rewards framework was limited to settings where the underlying RL agent was a planning (i.e., needs a model of the environment) agent that use lookahead search in some form (e.g, UCT). In these settings the only quantity being learned was the intrinsic reward function. By contrast, in this section we derive our algorithm for learning intrinsic rewards for the setting where the underlying RL agent is itself a learning agent, specifically a policy gradient based learning agent.

### 3.1   Policy Gradient based RL

Here we briefly describe how policy gradient based RL works, and then present our method that incorporates it. We assume an episodic, discrete-actions, RL setting. Within an episode, the state of

the environment at time step $t$ is denoted by $s_t \in S$ and the action the agent takes from action space $A$ at time step $t$ as $a_t$, and the reward at time step $t$ as $r_t$. The agent's policy, parameterized by $\theta$ (for example the weights of a neural network), maps a representation of states to a probability distribution over actions. The value of a policy $\pi_\theta$, denoted $J(\pi_\theta)$ or equivalently $J(\theta)$, is the expected discounted sum of rewards obtained by the agent when executing actions according to policy $\pi_\theta$, i.e.,

$$J(\theta) = E_{s_t \sim T(\cdot|s_{t-1},a_{t-1}), a_t \sim \pi_\theta(\cdot|s_t)}[\sum_{t=0}^{\infty} \gamma^t r_t], \tag{1}$$

where $T$ denotes the transition dynamics, and the initial state $s_0 \sim \mu$ is chosen from some distribution $\mu$ over states. Henceforth, for ease of notation we will write the above quantity as $J(\theta) = E_\theta[\sum_{t=0}^{\infty} \gamma^t r_t]$.

The policy gradient theorem of Sutton et al. [2000] shows that the gradient of the value $J$ with respect to the policy parameters $\theta$ can be computed as follows: from all time steps $t$ within an episode

$$\nabla_\theta J(\theta) = E_\theta[G(s_t, a_t)\nabla_\theta \log \pi_\theta(a_t|s_t)], \tag{2}$$

where $G(s_t, a_t) = \sum_{i=t}^{\infty} \gamma^{i-t} r_i$ is the return until termination. Note that recent advances such as advantage actor-critic (A2C) learn a critic ($V_\theta(s_t)$) and use it to reduce the variance of the gradient and bootstrap the value after every $n$ steps. However, we present this simple policy gradient formulation (Eq 2) in order to simplify the derivation of our proposed algorithm and aid understanding.

## 3.2 LIRPG: Learning Intrinsic Rewards for Policy Gradient

**Notation.** We use the following notation throughout.

- $\theta$: policy parameters
- $\eta$: intrinsic reward parameters
- $r^{ex}$: extrinsic reward from the environment
- $r_\eta^{in} = r_\eta^{in}(s, a)$: intrinsic reward estimated by $\eta$
- $G^{ex}(s_t, a_t) = \sum_{i=t}^{\infty} \gamma^{i-t} r_i^{ex}$
- $G^{in}(s_t, a_t) = \sum_{i=t}^{\infty} \gamma^{t-i} r_\eta^{in}(s_i, a_i)$
- $G^{ex+in}(s_t, a_t) = \sum_{i=t}^{\infty} \gamma^{i-t}(r_i^{ex} + \lambda r_\eta^{in}(s_i, a_i))$
- $J^{ex} = E_\theta[\sum_{t=0}^{\infty} \gamma^t r_t^{ex}]$
- $J^{in} = E_\theta[\sum_{t=0}^{\infty} \gamma^t r_\eta^{in}(s_t, a_t)]$
- $J^{ex+in} = E_\theta[\sum_{t=0}^{\infty} \gamma^t(r_t^{ex} + \lambda r_\eta^{in}(s_t, a_t))]$
- $\lambda$: relative weight of intrinsic reward.

The departure point of our approach to reward optimization for policy gradient is to distinguish between the extrinsic reward, $r^{ex}$, that defines the task, and a separate intrinsic reward $r^{in}$ that additively transforms the extrinsic reward and influences learning via policy gradients. It is crucial to note that the ultimate measure of performance we care about improving is

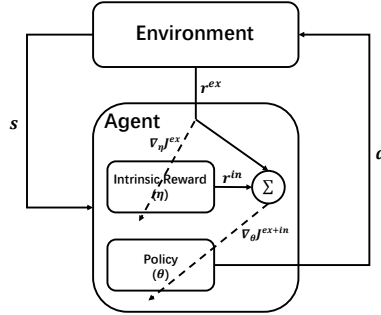

Figure 1: Inside the agent are two modules, a policy function parameterized by $\theta$ and an intrinsic reward function parameterized by $\eta$. In our experiments the policy function (A2C / PPO) has an associated value function as does the intrinsic reward function (see supplementary materials for details). As shown by the dashed lines, the policy module is trained to optimize the weighted sum of intrinsic and extrinsic rewards while the intrinsic reward module is trained to optimize just the extrinsic rewards.

the value of the extrinsic rewards achieved by the agent; the intrinsic rewards serve only to influence the change in policy parameters. Figure 1 shows an abstract representation of our intrinsic reward augmented policy gradient based learning agent.

**Algorithm Overview.** An overview of our algorithm, LIRPG, is presented in Algorithm 1. At each iteration of LIRPG, we simultaneously update the policy parameters $\theta$ and the intrinsic reward parameters $\eta$. More specifically, we first update $\theta$ in the direction of the gradient of $J^{ex+in}$ which is the weighted sum of intrinsic and extrinsic rewards. After updating policy parameters, we update $\eta$ in the direction of the gradient of $J^{ex}$ which is just the extrinsic rewards. Intuitively, the policy is updated to maximize the sum of extrinsic and intrinsic rewards, while the intrinsic reward function is updated to maximize only the extrinsic reward. We describe more details of each step below.

---
**Algorithm 1** LIRPG: Learning Intrinsic Reward for Policy Gradient
---
1: **Input:** step-size parameters $\alpha$ and $\beta$
2: **Init:** initialize $\theta$ and $\eta$ with random values
3: **repeat**
4:   Sample a trajectory $\mathcal{D} = \{s_0, a_0, s_1, a_1, \cdots\}$ by interacting with the environment using $\pi_\theta$
5:   Approximate $\nabla_\theta J^{ex+in}(\theta; \mathcal{D})$ by Equation 4
6:   Update $\theta' \leftarrow \theta + \alpha \nabla_\theta J^{ex+in}(\theta; \mathcal{D})$
7:   Approximate $\nabla_{\theta'} J^{ex}(\theta'; \mathcal{D})$ on $\mathcal{D}$ by Equation 11
8:   Approximate $\nabla_\eta \theta'$ by Equation 10
9:   Compute $\nabla_\eta J^{ex} = \nabla_{\theta'} J^{ex}(\theta'; \mathcal{D}) \nabla_\eta \theta'$
10:   Update $\eta' \leftarrow \eta + \beta \nabla_\eta J^{ex}$
11: **until** done
---

**Updating Policy Parameters ($\theta$).**   Given an episode where the behavior is generated according to policy $\pi_\theta$, we update the policy parameters using regular policy gradient using the sum of intrinsic and extrinsic rewards as the reward:

$$\theta' = \theta + \alpha \nabla_\theta J^{ex+in}(\theta) \tag{3}$$

$$\approx \theta + \alpha G^{ex+in}(s_t, a_t) \nabla_\theta \log \pi_\theta(a_t|s_t), \tag{4}$$

where Equation 4 is a stochastic gradient update.

**Updating Intrinsic Reward Parameters ($\eta$).**   Given an episode and the updated policy parameters $\theta'$, we update intrinsic reward parameters. Intuitively, updating $\eta$ requires estimating the effect such a change would have on the extrinsic value through the change in the policy parameters. Our key idea is to use the chain rule to compute the gradient as follows:

$$\nabla_\eta J^{ex} = \nabla_{\theta'} J^{ex} \nabla_\eta \theta', \tag{5}$$

where the first term ($\nabla_{\theta'} J^{ex}$) sampled as

$$\nabla_{\theta'} J^{ex} \approx G^{ex}(s_t, a_t) \nabla_{\theta'} \log \pi_{\theta'}(a_t|s_t) \tag{6}$$

is an approximate stochastic gradient of the extrinsic value with respect to the updated policy parameters $\theta'$ when the behavior is generated by $\pi_{\theta'}$, and the second term can be computed as follows:

$$\nabla_\eta \theta' = \nabla_\eta \left( \theta + \alpha G^{ex+in}(s_t, a_t) \nabla_\theta \log \pi_\theta(a_t|s_t) \right) \tag{7}$$

$$= \nabla_\eta \left( \alpha G^{ex+in}(s_t, a_t) \nabla_\theta \log \pi_\theta(a_t|s_t) \right) \tag{8}$$

$$= \nabla_\eta \left( \alpha \lambda G^{in}(s_t, a_t) \nabla_\theta \log \pi_\theta(a_t|s_t) \right) \tag{9}$$

$$= \alpha \lambda \sum_{i=t}^{\infty} \gamma^{i-t} \nabla_\eta r_\eta^{in}(s_i, a_i) \nabla_\theta \log \pi_\theta(a_t|s_t). \tag{10}$$

Note that to compute the gradient of the extrinsic value $J^{ex}$ with respect to the intrinsic reward parameters $\eta$, we needed a new episode with the updated policy parameters $\theta'$ (cf. Equation 6), thus requiring two episodes per iteration. To improve data efficiency we instead *reuse* the episode generated by the policy parameters $\theta$ at the start of the iteration and correct for the resulting mismatch by replacing the on-policy update in Equation 6 with the following off-policy update using importance sampling:

$$\nabla_{\theta'} J^{ex} = G^{ex}(s_t, a_t) \frac{\nabla_{\theta'} \pi_{\theta'}(a_t|s_t)}{\pi_\theta(a_t|s_t)}. \tag{11}$$

The parameters $\eta$ are updated using the product of Equations 10 and 11 with a step-size parameter $\beta$; this approximates a stochastic gradient update (cf. Equation 5).

**Implementation on A2C and PPO.**   We described LIRPG using the most basic policy gradient formulation for simplicity. There have been many advances in policy gradient methods that reduce the variance of the gradient and improve the data-efficiency. Our LIRPG algorithm is also compatible with such actor-critic architectures. Specifically, for our experiments on Atari games we used a

reasonably state of the art advantage actor-critic (A2C) architecture, and for our experiments on Mujoco domains we used a similarly reasonably state of the art proximal policy optimization (PPO) architecture. We provide all implementation details in supplementary material. [1]

# 4 Experiments on Atari Games

Our overall objective in the following first set of experiments is to evaluate whether augmenting a policy gradient based RL agent with intrinsic rewards learned using our LIRPG algorithm (henceforth, augmented agent in short) improves performance relative to the baseline policy gradient based RL agent that uses just the extrinsic reward (henceforth, A2C baseline agent in short). To this end, we first perform this evaluation on multiple Atari games from the Arcade Learning Environment (ALE) platform [Bellemare et al., 2013] using the same open-source implementation with exactly the same hyper-parameters of the A2C algorithm [Mnih et al., 2016] from OpenAI [Dhariwal et al., 2017] for both our augmented agent as well as the baseline agent. The extrinsic reward used is the game score change as is standard for the work on Atari games. The LIRPG algorithm has two additional parameters relative to the baseline algorithm, the parameter $\lambda$ that controls how the intrinsic reward is scaled before adding it to the extrinsic reward and the step-size $\beta$; we describe how we choose these parameters below in our results.

We also conducted experiments against two other baselines. The first baseline simply added a constant positive value as a live bonus to the agent's reward at each time step (henceforth, A2C-live-bonus baseline agent in short). The live bonus heuristic encourages the agent to live longer so that it will potentially have a better chance of getting extrinsic rewards. The second baseline augmented the agent with a count-based bonus generated by the pixel-SimHash algorithm [Tang et al., 2017] (henceforth, A2C-pixel-SimHash baseline agent in short.)

Note that the policy module inside the agent is really two networks, a policy network and a value function network (that helps estimate $G^{ex+in}$ as required in Equation 4). Similarly the intrinsic reward module in the agent is also two networks, a reward function network and a value function network (that helps estimate $G^{ex}$ as required in Equation 6).

## 4.1 Implementation Details

The intrinsic reward module has two very similar neural network architectures as the policy module described above. It has a "policy" network that instead of a softmax over actions produces a scalar reward for every action through a tanh nonlinearity to keep the scalar output in $[-1, 1]$; we will refer to it as the intrinsic reward network. It also has a value network that estimates $G^{ex}$; this has the same architecture as the intrinsic reward network except for the output layer that has a single scalar output without a non-linear activation. These two networks share the parameters of the first four layers with each other. We keep the default values of all hyper-parameters in the original OpenAI implementation of the A2C-based policy module unchanged for both the augmented and baseline agents. We use RMSProp to optimize the two networks of the intrinsic reward module. Recall that there are two parameters special to LIRPG. Of these, the step size $\beta$ was initialized to $0.0007$ and annealed linearly to zero over 50 million time steps for all the experiments reported below. We did a small hyper-parameter search for $\lambda$ for each game (described below).

## 4.2 Overall Performance

Figure 2 shows the improvements of the augmented agents over baseline agents on 15 Atari games: Alien, Amidar, Asterix, Atlantis, BeamRider, Breakout, DemonAttack, DoubleDunk, MsPacman, Qbert, Riverraid, RoadRunner, SpaceInvaders, Tennis, and UpNDown. We picked as many games as our computational resources allowed in which the published performance of the underlying A2C baseline agents was good but where the learning was not so fast in terms of sample complexity so as to leave little room for improvement. We ran each agent for 5 separate runs each for 50 million time steps on each game for both the baseline agents and augmented agents. For the augmented agents, we explored the following values for the intrinsic reward weighting coefficient $\lambda$, $\{0.003, 0.005, 0.01, 0.02, 0.03, 0.05\}$ and the following values for the term $\xi$, $\{0.001, 0.01, 0.1, 1\}$, that weights the loss from the value function estimates with the loss from the intrinsic reward function

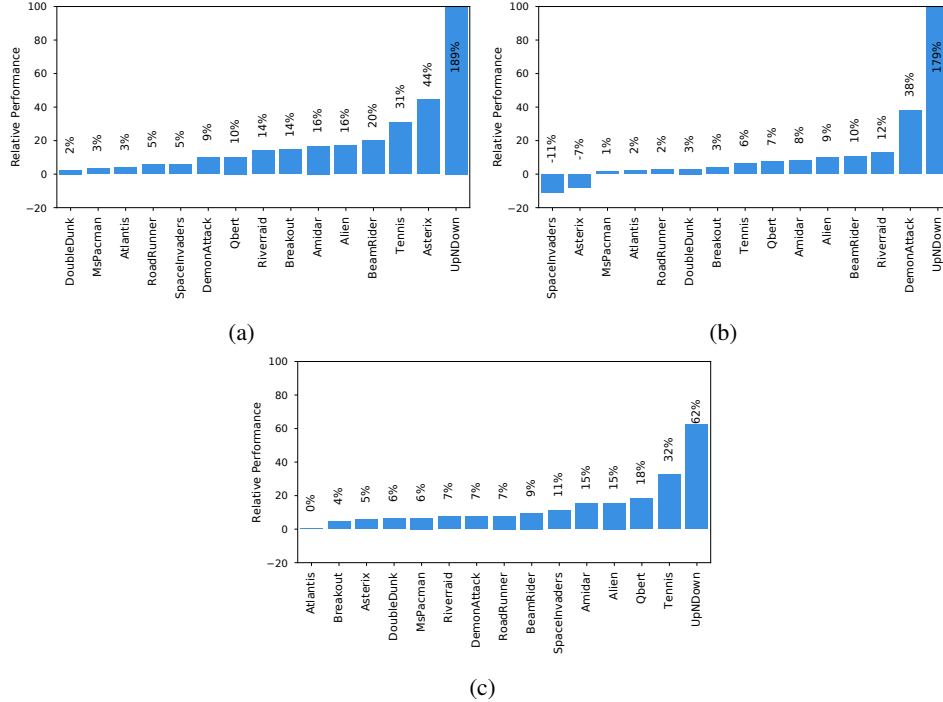

Figure 2: (a) Improvements of LIRPG augmented agents over A2C baseline agents. (b) Improvements of LIRPG augmented agents over live-bonus augmented A2C baseline agents. (c) Improvements of LIRPG augmented agents over pixel-SimHash augmented A2C baseline agents. In all figures, the columns correspond to different games labeled on the x-axes and the y-axes show human score normalized improvements.

(the policy component of the intrinsic reward module). We plotted the best results from the hyper-parameter search in Figure 2. For the A2C-live-bonus baseline agents, we explored the value of live bonus over the set $\{0.001, 0.01, 0.1, 1\}$ on two games, Amidar and MsPacman, and chose the best performing value of $0.01$ for all 15 games. For the A2C-pixel-SimHash baseline agents, we adopted all hyper-parameters from [Tang et al., 2017]. The learning curves of all agents are provided in the supplementary material.

The blue bars in Figure 2 show the human score normalized improvements of the augmented agents over the A2C baseline agents, the A2C-live-bonus baseline agents, and the A2C-pixel-SimHash baseline agents. We see that the augmented agent outperforms the A2C baseline agent on all 15 games and has an improvement of more than ten percent on 9 out of 15 games. As for the comparison to the A2C-live-bonus baseline agent, the augmented agent still performed better on all games except for SpaceInvaders and Asterix. Note that most Atari games are shooting games so the A2C-live-bonus baseline agent is expected to be a stronger baseline. The augmented agent outperformed or was comparable to the A2C-pixel-SimHash baseline agent on all 15 games.

### 4.3 Analysis of the Learned Intrinsic Reward

An interesting question is whether the learned intrinsic reward function learns a general state-independent bias over actions or whether it is an interesting function of state. To explore this question we used the learned intrinsic reward module and the policy module from the end of a good run (cf. Figure 2) for each game with no further learning to collect new data for each game. Figure 3 shows the variation in intrinsic rewards obtained and the actions selected by the agent over 100 thousand steps, i.e., 400 thousand frames, on 5 games. The analysis for all 15 games is in the supplementary material. The red bars show the average intrinsic reward per-step for each action. The black segments show the standard deviation of the intrinsic rewards. The blue bars show the frequency of each action being selected. Figure 3 shows that the intrinsic rewards for most actions vary through the episode as shown by large black segments, indirectly confirming that the intrinsic reward module learns more

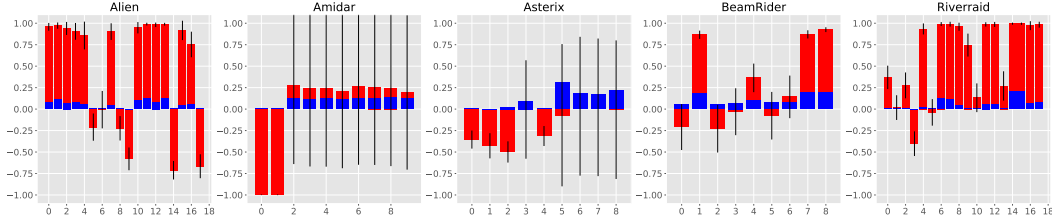

Figure 3: Intrinsic reward variation and frequency of action selection. For each game/plot the x-axis shows the index of the actions that are available in that game. The red bars show the means and standard deviations of the intrinsic rewards associated with each action. The blue bars show the frequency of each action being selected.

than a state-independent constant bias over actions. By comparing the red bars and the blue bars, we see the expected correlation between aggregate intrinsic reward over actions and their selection (through the policy module that trains on the weighted sum of extrinsic and intrinsic rewards).

## 5  Mujoco Experiments

Our main objective in the following experiments is to demonstrate that our LIRPG-based algorithm can extend to a different class of domains and a different choice of baseline actor-critic architecture (namely, PPO instead of A2C). Specifically, we explore domains from the Mujoco continuous control benchmark [Duan et al., 2016a], and used the open-source implementation of the PPO [Schulman et al., 2017] algorithm from OpenAI [Dhariwal et al., 2017] as our baseline agent. We also compared LIRPG to the simple heuristic of giving a live bonus as intrinsic reward (PPO-live-bonus baseline agents for short). As for the Atari game results above, we kept all hyper-parameters unchanged to default values for the policy module of both baseline and augmented agents. Finally, we also conduct a preliminary exploration into the question of how robust the learning of intrinsic rewards is to the sparsity of extrinsic rewards. Specifically, we used the *delayed* versions of the Mujoco domains, where the extrinsic reward is made sparse by accumulating the reward for $N = 10, 20, 40$ time steps before providing it to the agent. Note that the live bonus is not delayed when we delay the extrinsic reward for the PPO-live-bonus baseline agent. We expect that the problem becomes more challenging with increasing $N$ but expect that the learning of intrinsic rewards (that are available at every time step) can help mitigate some of that increasing hardness.

**Delayed Mujoco benchmark.**  We evaluated 5 environments from the Mujoco benchmark, i.e., Hopper, HalfCheetah, Walker2d, Ant, and Humanoid. As noted above, to create a more-challenging sparse-reward setting we accumulated rewards for $10, 20$ and $40$ steps (or until the end of the episode, whichever comes earlier) before giving it to the agent. We trained the baseline and augmented agents for 1 million steps on each environment.

### 5.1  Implementation Details

The intrinsic reward function networks are quite similar to the two networks in the policy module. Each network is a multi-layer perceptron (MLP) with 2 hidden layers. We concatenated the observation vector and the action vector as the input to the intrinsic reward network. The first two layers are fully connected layers with $64$ hidden units. Each hidden layer is followed by a tanh non-linearity. The output layer has one scalar output. We apply tanh on the output to bound the intrinsic reward to $[-1, 1]$. The value network to estimate $G^{ex}$ has the same architecture as the intrinsic reward network except for the output layer that has a single scalar output without a non-linear activation. These two networks do not share any parameters. We keep the default values of all hyper-parameters in the original OpenAI implementation of PPO unchanged for both the augmented and baseline agents. We use Adam [Kingma and Ba, 2014] to optimize the two networks of the intrinsic reward module. The step size $\beta$ was initialized to $0.0001$ and was fixed over 1 million time steps for all the experiments reported below. The mixing coefficient $\lambda$ was fixed to $1.0$ and instead we multiplied the extrinsic reward by $0.01$ cross all 5 environments.

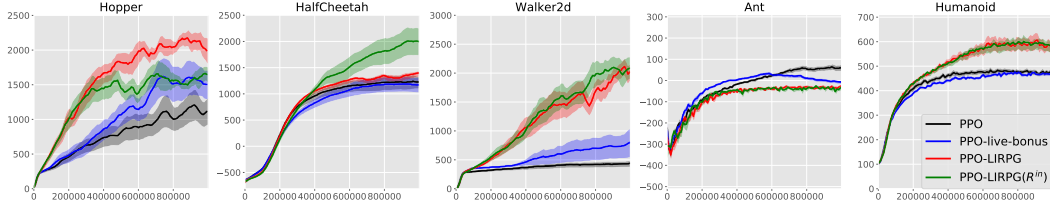

Figure 4: The x-axis is time steps during learning. The y-axis is the average reward over the last $100$ training episodes. The black curves are for the baseline PPO architecture. The blue curves are for the PPO-live-bonus baseline. The red curves are for our LIRPG based augmented architecture. The green curves are for our LIRPG architecture in which the policy module was trained with only intrinsic rewards. The dark curves are the average of $10$ runs with different random seeds. The shaded area shows the standard errors of $10$ runs.

## 5.2 Overall Performance

Our results comparing the use of learning intrinsic reward with using just extrinsic reward on top of a PPO architecture are shown in Figure 4. We only show the results of a delay of $20$ here; the full results can be found in the supplementary material. The black curves are for PPO baseline agents. The blue curves are PPO-live-bonus baseline agents, where we explored the value of live bonus over the set $\{0.001, 0.01, 0.1, 1\}$ and plotted the curves for the domain-specific best performing choice. The red curves are for the augmented LIRPG agents.

We see that in $4$ out of $5$ domains learning intrinsic rewards significantly improves the performance of PPO, while in one game (Ant) we got a degradation of performance. Although a live bonus did help on $2$ domains, i.e., Hopper and Walker2d, LIRPG still outperformed it on $4$ out of $5$ domains except for HalfCheetah on which LIRPG got comparable performance. We note that there was no domain-specific hyper-parameter optimization for the results in this figure; with such optimization there might be an opportunity to get improved performance for our method in all the domains.

**Training with Only Intrinsic Rewards.** We also conducted a more challenging experiment on Mujoco domains in which we used only intrinsic rewards to train the policy module. Recall that the intrinsic reward module is trained to optimize the extrinsic reward. In $3$ out of $5$ domains, as shown by the green curves denoted by 'PPO-LIRPG($R^{in}$)' in Figure 4, using only intrinsic rewards achieved similar performance to the red curves where we used a mixture of extrinsic rewards and intrinsic rewards. Using only intrinsic rewards to train the policy performed worse than using the mixture on Hopper but performed even better on HalfCheetah. It is important to note that training the policy using only live-bonus reward without the extrinsic reward would completely fail, because there would be no learning signal that encourages the agent to move forward. In contrast, our result shows that the agent can learn complex behaviors solely from the learned intrinsic reward on MuJoCo environment, and thus the intrinsic reward captures far more than a live bonus does; this is because the intrinsic reward module takes into account the extrinsic reward structure through its training.

## 6 Conclusion

Our experiments on using LIRPG with A2C on multiple Atari games showed that it helped improve learning performance in all of the $15$ games we tried. Similarly using LIRPG with PPO on multiple Mujoco domains showed that it helped improve learning performance in $4$ out $5$ domains (for the version with a delay of $20$). Note that we used the same A2C / PPO architecture and hyper-parameters in both our augmented and baseline agents. In summary, we derived a novel practical algorithm, LIRPG, for learning intrinsic reward functions in problems with high-dimensional observations for use with policy gradient based RL agents. This is the first such algorithm to the best of our knowledge. Our empirical results show promise in using intrinsic reward function learning as a kind of meta-learning to improve the performance of modern policy gradient architectures like A2C and PPO.

**Acknowledgments**

We thank Richard Lewis for conversations on optimal rewards. This work was supported by NSF grant IIS-1526059, by a grant from Toyota Research Institute (TRI), and by a grant from DARPA's L2M program. Any opinions, findings, conclusions, or recommendations expressed here are those of the authors and do not necessarily reflect the views of the sponsor.

## Footnotes

[1]Our implementation is available at: https://github.com/Hwhitetooth/lirpg

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
