[Supplementary Material · NIPS_2018_LIRPG_Camera_Ready_Supplementary_Material.pdf]

# Supplementary Material:
# On Learning Intrinsic Rewards for Policy Gradient Methods

**Zeyu Zheng**          **Junhyuk Oh**          **Satinder Singh**

Computer Science & Engineering
University of Michigan
{zeyu,junhyuk,baveja}@umich.edu

## A   Full Implementation Details

### A.1   Atari Experiments

**Episode Generation.**   As in Mnih et al. [2015], each episode starts by doing a no-op action for a random number of steps after restarting the game. The number of no-op steps is sampled from 0 to 30 uniformly. Within an episode, each action chosen is repeated for 4 frames, before selecting the next action. An episode ends when the game is over or the agent loses a life.

**Input State Representation.**   As in Mnih et al. [2015], we take the maximum value at each pixel from 4 consecutive frames to compress them into one frame which is then rescaled to a $84 \times 84$ gray scale image. The input to all four neural networks is the stack of the last 4 gray scale images (thus capturing frame-observations over 16 frames). The extrinsic rewards from the game are clipped to $[-1, 1]$.

**Details of the Two Networks in the Policy Module.**   Note that the policy module is unchanged from the OpenAI implementation. Specifically, the two networks are convolutional neural networks (CNN) with 3 convolutional layers and 1 fully connected layer. The first convolutional layer has thirty-two $8 \times 8$ filters with stride 4. The second convolutional layer has sixty-four $4 \times 4$ filters with stride 2. The third convolutional layer has sixty-four $3 \times 3$ filters with stride 1. The fourth layer is a fully connected layer with 512 hidden units. Each hidden layer is followed by a rectifier non-linearity. The value network (that estimates $G^{ex+in}$) shares parameters for the first four layers with the policy network. The policy network has a separate output layer with an output for every action through a softmax nonlinearity, while the value network separately outputs a single scalar for the value.

**Details of the Two Networks in the Intrinsic Reward Module.**   The intrinsic reward module has two very similar neural network architectures as the policy module described above. It again has two networks, a "policy" network that instead of a softmax over actions produces a scalar reward for every action through a tanh nonlinearity to keep the scalar in $[-1, 1]$; we will refer to it as the intrinsic reward function. The value network to estimate $G^{ex}$ has the same architecture as the intrinsic reward network except for the output layer that has a single scalar output without a non-linear activation. These two networks share the parameters of the first four layers.

**Hyper-Parameters for Policy module.**   We keep the default values of all hyper-parameters in the original OpenAI implementation of the A2C-based policy module unchanged for both the augmented and baseline agents[1].

**Hyper-Parameters for Intrinsic Reward module in Augmented Agent.** We use RMSProp to optimize the two networks of the intrinsic reward module. The decay factor used for RMSProp is 0.99, and the $\epsilon$ is 0.00001. We do not use momentum. Recall that there are two parameters special to LIRPG. Of these, the step size $\beta$ was initialized to 0.0007 and annealed linearly to zero over 50 million time steps for all the experiments reported below. We did a small hyper-parameter search for $\lambda$ for each game (this is described below in the caption of Figure 1). As for the A2C implementation for the policy module we clipped the gradient by norm to 0.5 in the intrinsic reward module.

## A.2 Mujoco Experiments

**Details of the Two Networks in the Policy Module.** Note that the policy module is unchanged from the OpenAI implementation; we provide details for completeness. The policy network is a MLP with 2 hidden layers, too. The input to the policy network is the observation. The first two layer are fully connected layers with 64 hidden units. Each hidden layer is followed by a tanh non-linearity. The output layer outputs a vector with the size of the dimension of the action space with no non-linearity applied to the output units. Gaussian noise is added to the output of the policy network to encourage exploration. The variance of the Gaussian noise was a input-independent parameter which was also trained by gradient descent. The corresponding value network (that estimates $G^{ex+in}$) has a similar architecture with the policy network. The only difference is that that output layer outputs a single scalar without any non-linear activation. These two networks do not share any parameters.

**Details of the Two Networks in the Intrinsic Reward Module.** The intrinsic reward function networks are quite similar to the two networks in the policy module. Each network is a multi-layer perceptron (MLP) with 2 hidden layers. We concatenated the observation vector and the action vector as the input to the intrinsic reward network. The first two layer are fully connected layers with 64 hidden units. Each hidden layer is followed by a tanh non-linearity. The output layer has one scalar output. We apply tanh on the output to bound the intrinsic reward to $[-1, 1]$. The value network to estimate $G^{ex}$ has the same architecture as the intrinsic reward network except for the output layer that has a single scalar output without a non-linear activation. These two networks do not share any parameters.

**Hyper-Parameters for Policy Module** We keep the default values of all hyper-parameters in the original OpenAI implementation of PPO unchanged for both the augmented and baseline agents[2].

**Hyper-Parameters for Intrinsic Reward Module** We use Adam [Kingma and Ba, 2014] to optimize the two networks of the intrinsic reward module. The step size $\beta$ was initialized to 0.0001 and was fixed over 1 million time steps for all the experiments reported below. The mixing coefficient $\lambda$ was fixed to 1.0 and instead we multiplied the extrinsic reward by 0.01 cross all 5 environments. The PPO implementation clips the gradient by norm to 0.5. We keep this part unchanged for the policy network and clip the gradients by the same norm for the reward network. We used generalized advantage estimate (GAE) [Schulman et al., 2015] for both training the reward network and the policy network. The weighting factor for GAE was 0.95.

# B More Experimental Results

---

entropy regularization term in the objective function are 1.0, 0.5, and 0.01. The learning rate $\alpha$ for training the policy is set to 0.0007 at the beginning and anneals to 0 linearly over 50 million steps. The discount factor $\gamma$ is 0.99 for all experiments.

[2]For each training iteration, the agent interacts with the environment for 2048 steps. The learning rate $\alpha$ for training the policy is set to 0.0003 at the beginning and was fixed over training. We used a batch size of 32 and swept over the 2048 data points for 10 epochs before the next sequence of interaction. The discount factor $\gamma$ is 0.99 for all experiments.

Figure 1: The x-axis is time steps during learning. The y-axis is the average game score over the last 100 training episodes. The black curves are for the baseline architecture. The deep blue curves are for the A2C-live-bonus baseline. The light blue curves are for the A2C-pixel-SimHash baseline. The red curves are for our LIRPG based augmented architecture. The dark curves are the average of four runs with different random seeds. The shaded areas show the standard errors of 5 individual runs. *Hyper-parameter Search:* We explored the following values for the intrinsic reward weighting coefficient $\lambda$, $\{0.003, 0.005, 0.01, 0.02, 0.03, 0.05\}$. We explored the following values for the term $\xi$, $\{0.001, 0.01, 0.1, 1\}$, that weights the loss from the value function estimates with the loss from the intrinsic reward function (the policy component of the intrinsic reward module).

Figure 2: Intrinsic reward variation and frequency of action selection. We selected a good run for each game from the runs shown in Figure 1, and used the learned intrinsic reward module and the associated policy module for the selected run without any further learning to play the game for 100 thousand steps, i.e., 400 thousand frames, to collect data. For each game/plot the x-axis shows the index of the actions that are available in that game. The red bars show the means and standard deviations of the intrinsic rewards associated with each action. The blue bars show the frequency of each action being selected.

Figure 3: The x-axis is time steps during learning. The y-axis is the average reward over the last 100 training episodes. Each column corresponds to a domain labeled at the top. Each row corresponds to the delay labeled on the left hand side (for 10, 20, and 40 steps from the top row to the bottom row). The black curves are for the baseline PPO architecture. The blue curves are for the PPO-live-bonus baseline. The red curves are for our LIRPG based augmented architecture. The green curves are for our LIRPG architecture in which the policy module was trained with only intrinsic rewards. The dark curves are the average of 10 runs with different random seeds. The shaded area shows the standard errors of 10 runs.

## Footnotes

[1]We use 16 actor threads to generate episodes. For each training iteration, each actor acts for 5 time steps. For training the policy, the weighting coefficients of policy-gradient term, value network loss term, and the