[Reviews · NeurIPS 2018]

Reviewer 1



This work attempts to use learnable intrinsic rewards in addition to conventional extrinsic reward from the environment to boost the agent performance calculated as conventional returns as the cumulative extrinsic rewards. This work can be seen as a variant of previous reward shaping and auxiliary rewards works but cannot be a more general version because though with more general mathematical form, it loses the consideration of domain knowledges, the key insights of previous works. Compared with its closest related works [Sorg et al. 2010, Guo et al. 2016], the method proposed here can be used in bootstrapping learning agents rather than only planning agents (i.e., Monte Carlo sampling of returns). Strengths: 1. This work implements an intuitively interesting idea that automatically shaping the rewards to boost learning performance. And compared with closely related previous works, it is more general to be used in modern, well performed learning methods (e.g., A2C and PPO). 2. The presentation is clear and easy to follow. 3. The experiment evaluation shows obvious improvement in the performance of the proposed method. Weakness: 1. The technical design is in lack of theoretical support. Even if we accept the design that the policy parameters are updated to optimize the combinational returns while the intrinsic reward parameters are updated to optimize only the extrinsic returns, which itself needs more justification (though has been taken in Sorg et al. 2010), the update derivation for the intrinsic reward parameters (i.e., Eq.7-10) is hardly convincing. 2. Though the superior performance is shown in the experiment, there are still worse performance cases (e.g., the forth panel in Fig.4), which lacks explanation. 3. As pointed out in Sec.1, the basic assumption to use reward shaping (or adding intrinsic reward) is to "change for better or for worse the sample (and computational) complexity of the RL agent learning from experience in its environment using the transformed reward function". But throughout the work, no theoretical, or experimental analysis can support this assumption based on the proposed design. One possible way to repair it will be to use simple thought/simulation experiments to showcase this, which will be much clearer than the performance shown in complicated benchmarks.

Reviewer 2



This paper proposes an algorithm for learning parametric intrinsic rewards for policy-gradient based learning which is built on optimal rewards framework (Singh et al. [2010]). The main part of paper (Sec 3.2) is about how to calculate the gradients and does parameter updates. Since their method needs two episodes to calculate the gradients and update parameters, samples from the previous episode are used in order to improve data efficiency. Even though this paper addresses the deficiency of reward function in RL with learnable ones, I am not completely convinced about the contribution and novelty of this paper. Becuase parametric intrinsic rewards have been studied before [Jaderberg et al., 2016, Pathak 2017, Singh et al. 2010] in both policy and value-based methods. I might be missing something here but this paper only shows how to derive the gradients and it doesn't introduce any new type of intrinsic reward function or learning?! The experiments sections contain a various comparison with policy gradient methods. The proposed model shows improvement over using policy gradient methods without any intrinsic reward, but comparisons with a method that uses a form of intrinsic reward would have been very informative. The writing is not clear in some parts of paper and needs to be improved: - Lines 30-34: it is very unclear and lots of terms used without proper definition, e.g. potential-based reward or order-preserving reward - Lines 38-40: ' Thus, in addressing the reward-design problem one may want to consider ... ' is unclear to me Question: What were the criteria for selecting 15 games among all others for the experiments?

Reviewer 3



This paper derives a novel algorithm for learning intrinsic rewards for policy-gradient based learning agents. The intrinsic reward definition is based on the Optimal Rewards Framework of Singh et al. The learning is developed into two phases. In the first phase, the algorithm updates the policy parameter to maximize the sum of intrinsic reward and the extrinsic reward. In the second phase, the algorithm updates the intrinsic reward parameters to maximize only the extrinsic reward. The overall effect is to find the intrinsic reward that helps us to maximize the extrinsic reward faster. The authors did an extensive experiment on Atari games and Mujoco domains, and showed improved performance on most of the tasks. The paper is well written and easy to follow. The experiment is extensive and solid. The idea is simple and interesting. I have only several minor comments. Sec 3.1. It is nice to mention the REINFOCE algorithm in the related work. Sec 3.2. It is more clear to mention that lambda is a hyper parameter to be tuned. I spent some time wondering if lambda is a parameter to be learned by the algorithm. Equ 4. Do you need to sum t from 1 to T? Equ 5. The notation need to be improved. Here, you are actually taking derivative over J^ex(theta’) but not J^ex ( J^ex is defined on line 152) Equ 11. Will the gradient becomes too large due to importance sampling?